# Influence of Short-Pulse Microwave Radiation on Thermochemical Properties Aluminum Micropowder

**DOI:** 10.3390/ma16030951

**Published:** 2023-01-19

**Authors:** Andrei Mostovshchikov, Fedor Gubarev, Olga Nazarenko, Alexey Pestryakov

**Affiliations:** 1School of Core Engineering Education, Tomsk Polytechnic University, Tomsk 634050, Russia; 2Department of Physical Electronics, Tomsk State University of Control Systems and Radioelectronics, Tomsk 634034, Russia; 3Institute of Nuclear Energy and Industry, Sevastopol State University, Sevastopol 299053, Russia; 4School of Non-Destructive Testing, Tomsk Polytechnic University, Tomsk 634050, Russia; 5Research School of Chemistry and Applied Biomedical Sciences, Tomsk Polytechnic University, Tomsk 634050, Russia

**Keywords:** microwave radiation, metal powder, thermal analysis

## Abstract

The thermochemical properties of Al micropowder after exposure to microwave irradiation were investigated. The Al micropowder was exposed to microwave irradiation in air with a frequency of 2.85 GHz, a power density of 8 W/cm^2^, and a pulse duration of 25 ns and 3 µs. The thermochemical parameters of the irradiated metal powders were determined by the method of thermal analysis at the heating in air. It was found that an increase in the duration of microwave pulses and irradiation time leads to the thermal annealing of the metal particles, and the thermal processes of melting and sintering begin to dominate over non-thermal processes. The specific thermal effect of irradiated Al micropowder oxidation increases from 7744 J/g to 10,154 J/g in comparison with the unirradiated powder. The modeling of thermal heating processes of aluminum (Al) micropowder under the action of pulsed microwave radiation has been performed. It is shown that with an increase in the duration of microwave pulses and irradiation time, a significant heating of the Al micropowder occurs, leading to its melting and sintering. The results of modeling on the action of microwave radiation on the Al micropowder were compared with experimental results.

## 1. Introduction

Metal powders are used in a wide range of industrial applications including 3D printing technologies [1], additive technologies [2], pyrotechnics [3], and ceramics production [4]. Increasing the reactivity of metal powders in various processes contributes to lower energy costs and saving resources [5]. At present, in order to give the metal powders the required properties, they are modified mainly by introducing various chemical additives into them [6,7], by changing their dispersity by mechanical activation [8], or the particle shape by spheroidization [9]. The significant disadvantages of such methods of powder modification are the contamination of the original powder with other chemicals, or a change in the shape of the particles and the distribution of powder particles by diameter. Recently, a number of works have been published aimed at studying, and the technological application of continuous microwave radiation for the melting and sintering of metal powders [10,11,12]. The exposure of various materials to microwaves is accompanied by dielectric heating and resonance absorption due to rotational excitation, while microwave heating of metals occurs predominantly due to the magnetic field based loss effects [11,13]. In this regard, in order to simulate thermal processes in metal powders when exposed to microwave radiation, a mathematical apparatus was developed, which is described in detail in the works [10,14]. In addition, it was experimentally found that under the action of short-pulse microwave radiation, metal powders change their thermochemical properties [15,16]. The temperature at the beginning of the oxidation of aluminum powder decreases, the specific heat of oxidation increases, and the degree of oxidation when heated in air increases after exposure to microwave radiation with a frequency of 9.4 GHz and a power density of 80 W/cm^2^ [15]. On the other hand, it was shown that with an increase in the duration of microwave pulses and irradiation time, the thermal heating and annealing of the protective oxide shell on the surface of aluminum particles occur, which leads to a decrease in the specific thermal effect of oxidation. Moreover, the physical processes of non-thermal nature in micro- and nanopowders of aluminum and iron were observed, and the difference in the processes under the action of microwave irradiation on powders was explained by the different structure of the oxide shells of particles of aluminum and iron powders [16]. Thus, the use of short-pulse microwave radiation is a promising direction for the purposeful modification of metal powders without the introduction of any additives and without changing the dispersion.

For the further development of the method for modifying powders by short-pulse radiation, a theoretical assessment of the thermal heating of the powder is a fundamentally important problem. In this case, it is necessary to determine the regimes of pulsed action under which there is no significant heating of the powder, leading to its melting and sintering. This will make it possible to effectively use pulsed microwave radiation for modifying metal powders by non-thermal radiation interaction mechanism.

This work aimed to carry out a theoretical estimation of the thermal effect of pulsed microwave radiation on aluminum micropowder and determine the boundary duration of a microwave radiation pulse, less than which it is still possible for non-thermal processes to take place in the irradiated powder. One more goal was to verify the theoretical estimate of the microwave pulse duration, at which the transition from non-thermal processes in the irradiated powders to thermal ones occurs.

## 2. Materials and Methods

### 2.1. Materials

For the experimental study of the microwave radiation effect on the thermochemical parameters of metal powders, the industrial micron aluminum (Al) powder obtained by the melt spray [17] was chosen. Figure 1 shows scanning electron microscopy (SEM) images of the studied metal powders.

The particles of Al micropowder are not sintered and coated with a stable oxide shell consisting only of a monovalent oxide [18]. The mean surface diameter of particles is 6.5 μm.

### 2.2. Method

The experimental setup for studying the effect of microwave radiation on the properties of metal powders is shown in Figure 2. It consisted of the following parts: the microwave radiation source (1), which based on the magnetron oscillator; the antenna-feeder system, which included ferrite valve (2), directional coupler (3), waveguide channel (5); the horn antenna (7); the anechoic chamber (6) with the absorption coefficient of electromagnetic radiation 26–27 dB; the oscilloscope DS 091204 A (4) for control and measurement of pulse parameters. The object of research was a thin layer of Al micropowder (thickness of ~2 mm) (8) located on the dielectric substrate (9) made of aluminum oxide. Aluminum oxide is transparent material for short-pulse microwave radiation. The layer of Al micropowder was placed in the working area of the chamber at a distance of up to 0.5 m from the horn antenna.

The PO-1 meter (PO Box M-5664, Nizhny Novgorod, Russia) was used to measure the power flux density in this work. The oscilloscope DS 091204 A (Agilent Technologies, Inc., Santa Clara, CA, USA) was used for measurements of pulse duration. The surface temperature of the powder samples was monitored using a thermal imager Fluke TiR10 (Fluke Corporation, Everett, WA, USA). The samples were treated with microwave radiation in two modes, the parameters of which are presented in Table 1. The duration of microwave irradiation was 5 s.

### 2.3. Characterization

The irradiated samples of metal powders were characterized by studying their thermochemical parameters [19,20]: the oxidation onset temperature T_ox_ (°C); specific thermal effect of oxidation ΔH (kJ/mol); the degree of oxidation α (%). The differential thermal analysis (DTA) was performed with a heating rate of 10 °C/min, from 20 to 1200 °C in air using SDT Q600 thermogravimetric analyzer (TA Instruments, New Castle, DE, USA).

## 3. Estimation of Thermal Effects of Short-Pulse Microwave Radiation

The estimation of the thermal effect of pulsed microwave radiation on Al powder was performed for the case of a plane electromagnetic wave incident on a powder layer, and heating this layer with it. The amplitude of the incident wave in the calculation corresponded to the power flux density in the experiments (80 W/cm^2^ for 9.4 GHz (X-band) radiation and 8 kW/cm^2^ for 2.85 GHz (S-band) radiation). Al micropowder with the following parameters was chosen as a model powder: mean surface particle diameter *d* = 6.5 µm, dielectric layer thickness *l* = 0.03 µm, and particle volume fraction *p* = 0.6.

The thickness of the powder layer was taken to be ~λ/4 (0.44 and 1.27 mm for the frequency of 9.4 and 2.85 GHz, respectively), so that the intensity profile along the layer of irradiated powder was approximately uniform,

The electromagnetic wavelength is much larger than the average particle diameter of Al micropowder (~6.5 μm), as a result the wave can penetrate much deeper into the powder layer than in the case of bulk aluminum metal. Thus, the aluminum powder layer can be considered as a dielectric layer with average values of the magnetic and dielectric permittivity *μ_eff_* and *ε_eff_*. The effective medium method was used to calculate these values, used to describe the macroscopic properties of composite materials from metal powders [10]. For instance, according to this method, the value of *ε_eff_* is defined as follows:(1)$$\langle D\rangle =\frac{1}{V}\int_{V}^{}D\left(r\right)dr=\frac{1}{V}\int_{V}^{}\varepsilon \left(r\right)E\left(r\right)dr=\varepsilon_{eff}E_{0}.$$
where 〈D〉 is electrical induction averaged over the volume *V* of the particle, $E_{0}$ is external electric field strength, $D\left(r\right)$, $E\left(r\right)$, $\varepsilon \left(r\right)$ are the local values of the electrical induction, electric field strength and dielectric constant.

To calculate the effective parameters of metal powders, expressions that take into account the effect of the oxide film on the parameters were taken [21]:(2)$$p\frac{\mu_{p}-\mu_{eff}}{\mu_{p}+2\mu_{eff}}+(1-p)\frac{\mu_{g}-\mu_{eff}}{\mu_{g}+2\mu_{eff}}=0$$
(3)$$p\frac{\varepsilon_{p}-\varepsilon_{eff}}{\varepsilon_{p}+2\varepsilon_{eff}}+(1-p)\frac{\varepsilon_{g}-\varepsilon_{eff}}{\varepsilon_{g}+2\varepsilon_{eff}}=0$$

The values of *ε_eff_* and *μ_eff_* are found by solving Equations (2) and (3), where
(4)$$\mu_{p}=\mu_{0}F_{0},\;\;\;\varepsilon_{p}=\varepsilon_{2}F_{2}^{\varepsilon },\;$$
(5)$$F_{2}^{\varepsilon }=2\frac{1-{\left(r_{1}/r_{2}\right)}^{3}F_{1}^{\varepsilon }}{2+{\left(r_{1}/r_{2}\right)}^{3}F_{1}^{\varepsilon }},\;\;\;F_{1}^{\varepsilon }=2\frac{1-(\varepsilon_{1}/\varepsilon_{2})F_{0}}{2+(\varepsilon_{1}/\varepsilon_{2})F_{0}}$$
(6)$$F_{0}(y)=2\frac{-y\cos y+\sin y}{y\cos y-\sin y+y^{2}\sin y}$$

For highly conductive non-magnetic metals:(7)$$y=(1+i)r_{1}/\delta ,\;\;\;\delta =\sqrt{\frac{2}{\omega \sigma \mu_{0}}}$$
(8)$$\varepsilon_{1}=i\frac{\sigma }{\varepsilon_{0}\omega }$$

Here *r*_1_ is the radius of the metal core of the particle, *r*_2_ is the particle radius, *ε*_1_ and *ε*_2_ are the dielectric permeability of the metal core and oxide film, *ε_g_* and *µ_g_* are the dielectric and magnetic permeability of the powder gas medium, *δ* is the skin depth of the metal core, *ε*_0_ and µ_0_ are the electrical and magnetic constants, *σ* is the electrical conductivity of the metal core, *ω* is the frequency of the radiation, *i* is the imaginary unit. According to the # 7 equation, the calculated skin depth *δ* for the Al particles at given frequencies are 3.84 μm (2.85 GHz) and 2.09 μm (9.4 GHz), respectively. The dielectric permeability for the oxide film of the Al particles was taken as *ε*_2_ = 10 + 0.01*i* [22].

The calculated effective parameters for the Al powder at given frequencies are shown in Table 2.

Table 2 shows that the greatest contribution to the heating of the medium is made by the electric component of the field. However, according to the data in Table 2, the values of the coefficient *ε_eff_* for the frequencies of 2.85 GHz and 9.4 GHz are almost the same. Therefore, the treatment of Al micropowder by microwave radiation of these frequency ranges will lead to the same intensity of heating of the powder.

Using the finite-difference method, the temperature distribution along the powder layer was calculated after irradiation with two types of pulses: 9.4 GHz with a duration of 3 µs (Figure 3a) and 2.85 GHz with a duration of 25 ns (Figure 4a). The temperature boundary condition was taken to be *T* = 20 °C. The heat distribution along the powder layer is calculated from the following expressions:(9)$$Q(z)=Q_{e}(z)+Q_{h}(z)=\frac{\omega }{8\pi }({\varepsilon_{eff}}^{\prime \prime }{\left|e(z)\right|}^{2}+{\mu_{eff}}^{\prime \prime }{\left|h(z)\right|}^{2})$$
(10)$$h(z)=h_{1}\exp(-i\omega t+ikz)+h_{2}\exp(-i\omega t-ik(z-d))$$
(11)$$e(z)=\frac{\mu_{eff}\omega }{ck}h_{1}\exp(-i\omega t+ikz)+\frac{\mu_{eff}\omega }{ck}h_{2}\exp(-i\omega t-ik(z-d))$$
(12)$$h_{1}=\frac{2h_{0}(1+\mu_{eff}\omega /ck)}{{\left(1+\mu_{eff}\omega /ck\right)}^{2}-{\left(1-\mu_{eff}\omega /ck\right)}^{2}\exp(2ikd)}$$
(13)$$h_{2}=-\frac{2h_{0}(1-\mu_{eff}\omega /ck)\exp(ikd)}{{\left(1+\mu_{eff}\omega /ck\right)}^{2}-{\left(1-\mu_{eff}\omega /ck\right)}^{2}\exp(2ikd)}$$
(14)$$k=\frac{\omega }{c}\sqrt{\varepsilon_{eff}\mu_{eff}}$$

It can be seen from Figure 3b and Figure 4b that the pulse duration should not exceed 150 μs for the X-band (9.4 GHz) and 3 μs for the S-band (2.85 GHz). With a longer pulse duration, the powder temperature will reach the aluminum melting point and will lead to powder sintering. Thus, these values of the pulse length are the boundary values of the duration microwave pulses at which nonthermal processes take place. Obviously, the processes of non-thermal nature in the powder particles can occur with a longer pulse length, but due to the heating, melting, and sintering of the powder the processes of non-thermal nature will not appear.

Figure 3c and Figure 4c show the characteristic cooling time of the Al powder after exposure to microwave radiation; it does not exceed several seconds.

## 4. Thermal Analysis Results

To verify the theoretical estimate of the microwave pulse duration, at which the transition from non-thermal processes in the irradiated powders to thermal ones occurs, the Al powder was experimentally treated with microwave radiation with the frequency of 2.85 GHz and the pulse duration of 25 ns and 3 μs.

Figure 5 shows the thermogram of Al micropowder after exposure to microwave radiation with the frequency of 2.85 GHz and the pulses duration of 25 ns.

According to the results of thermal analysis, after exposure to microwave irradiation with a pulse duration of 25 ns, the specific thermal effect of Al micropowder oxidation increases by 31.1 ± 1.8% (up to 10,154 J/g) in comparison with the unirradiated powder. In this case, the onset temperature of powder oxidation decreases to ~346 °C, and the content of sorbed water slightly increases (up to ~1.5%). A similar result was obtained when the Al powder was irradiated with microwave radiation at the frequency of 9.4 GHz and the pulse duration of 3 μs [16]. The increase in the specific thermal effect of oxidation in [16] was explained by the increase in the permeability of the oxide shell to the oxidizer due to ionization processes in the particle core under the action of microwave radiation and its electrostatic charging. There was also a partial reduction of the metal in the oxide shell, the dissociation of water on the particle surface, and the formation of a double electric layer with pseudocapacitance.

Figure 6 shows the thermogram of the Al micropowder after microwave irradiation with the frequency of 2.85 GHz and the pulse duration of 3 µs. According to the theoretical estimation, at this pulse duration, the thermal effect of microwave radiation is significantly appeared in the powder.

According to thermal analysis data, the increase in the specific thermal effect of oxidation (from 7744 J/g to 7872 J/g) of the Al micropowder was less than the error of the method (less than 1.6%). The onset temperature of oxidation decreased by 20 °C (to ~460 °C) in comparison with that of the unirradiated powder. Thus, this indicates that the processes of aluminum reduction in the oxide shell began to occur in the powder. At the same time, there was no significant decrease in the onset temperature of oxidation, probably due to the beginning of the powder thermal annealing and the accompanying oxidation of the reducing aluminum. In addition, the amount of sorbed water decreased to 0.8 wt%, which also indicates the beginning of thermal annealing of the powder.

It is believed that electroexplosive metal nanopowders, due to the high nonequilibrium of their production processes, contain a certain amount of the so-called “excess energy”, which some researchers also call “stored energy” [23,24]. In [24], the authors show that the stored energy effect is a thermodynamically highly nonequilibrium state of a material, as a result of which its chemical activity can vary by many orders of magnitude. The authors [24] also conclude that, in addition to the “stored energy” associated with nonequilibrium conditions for obtaining nanopowders and the high curvature of their surface, they can additionally store energy due to various defects using hard high-energy effects (neutron irradiation, gamma-ray radiation). The mechanisms of stabilization of a part of the radiation energy in a substance and its subsequent relaxation in the form of “excess” or “stored energy” require further study. In [25,26], we studied the effect of synchrotron radiation and the effect of an electron beam on the thermochemical characteristics of aluminum nanopowder, and showed a significant increase in the thermal effect.

## 5. Conclusions

According to theoretical estimates, there is no significant thermal heating of the Al micropowder when it is exposed to microwave radiation with a frequency of 2.85 GHz and a pulse duration of 25 ns. Consequently, the powder annealing does not occur during irradiation. Under microwave radiation with such short pulse duration, only non-thermal processes may occur. At the pulse duration of 3 µs, the effect of heating the powder by microwave radiation begins to appear in Al micropowder, thus, thermal annealing of the powder begins. This result is in good agreement with the experimental data.

According to the experimental results of our previous work, at the pulse duration of 3 μs and the microwave radiation frequency of 9.4 GHz, the thermal effect of Al micropowder increases. There is no significant heating in this exposure mode, according to the theoretical estimations carried out in this paper. Based on the comparison of the estimation results with the experimental results on the effect of microwave radiation on Al powder obtained in this work, as well as those obtained earlier, short-pulsed irradiation is most effective for modifying the physicochemical properties of micron aluminum powder. With an increase in the duration of microwave pulses and irradiation time, thermal annealing of aluminum particles occurs, and the thermal processes of melting and sintering begin to dominate over non-thermal processes, which was found experimentally in this work and explains the previously obtained results.

The results obtained in this work show that the use of pulsed microwave radiation, similarly to the use of the electron beam and synchrotron radiation, makes it possible to solve the problem of modifying dispersed metals to give them new functional properties—energy storage and changes in the structure of the passivating shell. The stored energy in irradiated metal powders makes it possible to increase the energy of inorganic synthesis processes by combustion, sintering, oxidation, and combustion processes, due to an additional exothermic effect associated with the release of stored energy.

## Figures and Tables

**Figure 1 materials-16-00951-f001:**
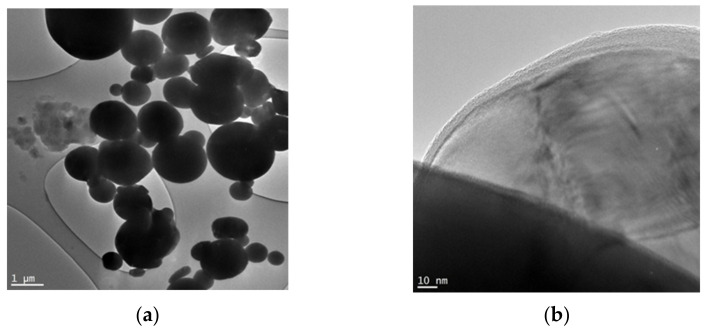
SEM images of Al powder: (**a**) General view, (**b**) High magnification.

**Figure 2 materials-16-00951-f002:**
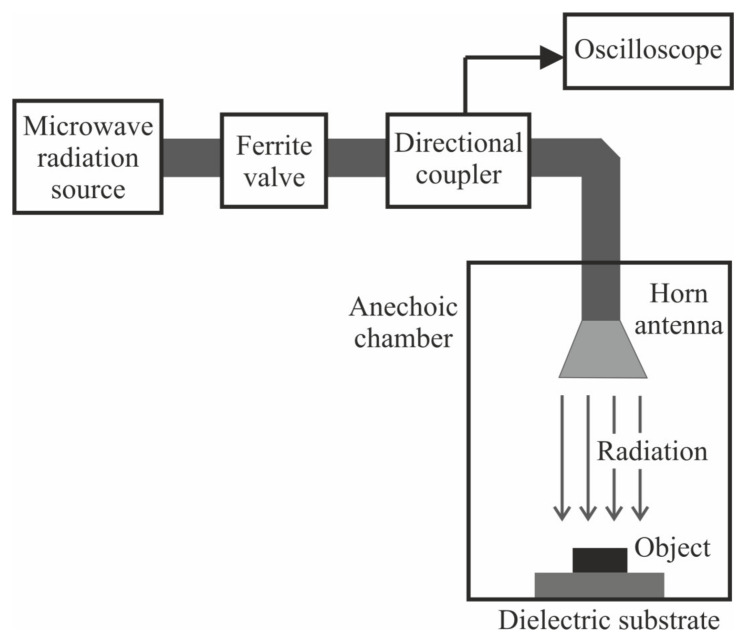
Scheme of an experimental setup for studying the effect of microwave radiation on metal powders.

**Figure 3 materials-16-00951-f003:**
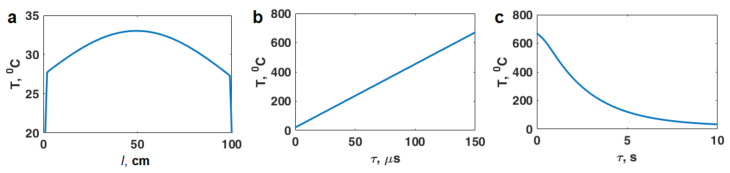
Thermal effect for microwave radiation of 9.4 GHz (X-band): (**a**) temperature distribution along the quarter-wave layer of the powder; (**b**) temperature dependence on the pulse duration; (**c**) temperature cooling dependence on time under the absence of irradiation.

**Figure 4 materials-16-00951-f004:**
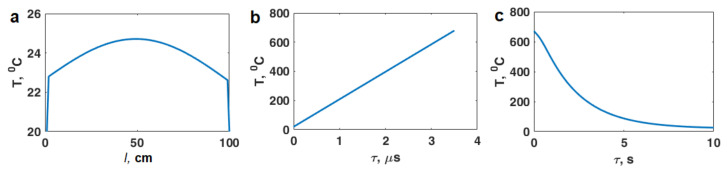
Thermal effect for microwave radiation of 2.85 GHz (S-band): (**a**) temperature distribution along the quarter-wave layer of the powder; (**b**) temperature dependence on the pulse duration; (**c**) temperature cooling dependence on time under the absence of irradiation.

**Figure 5 materials-16-00951-f005:**
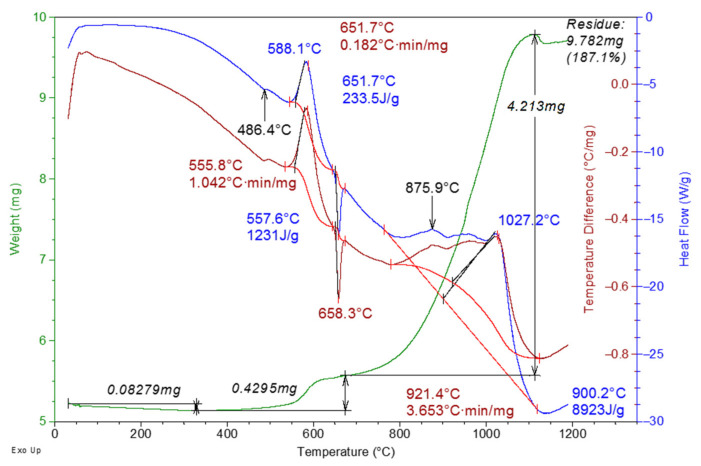
Thermogram of Al micropowder after exposure to microwave with a pulse duration of 25 ns.

**Figure 6 materials-16-00951-f006:**
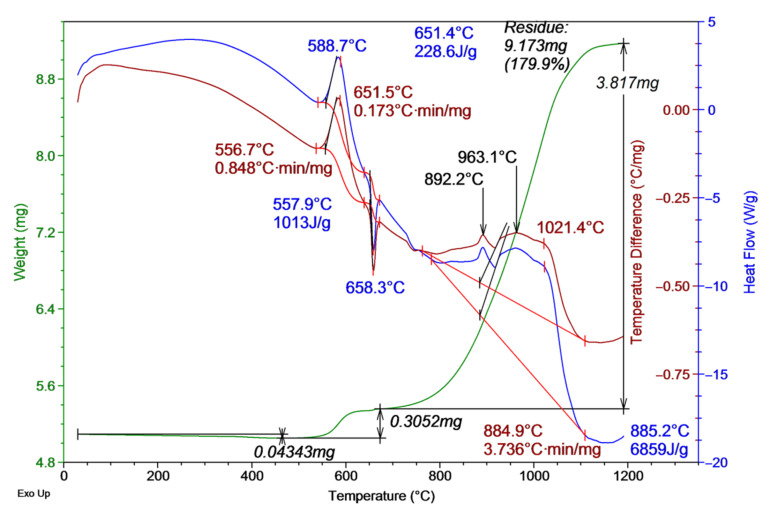
Thermogram of Al micropowder after exposure to microwave radiation with a pulse duration of 3 µs.

**Table 1 materials-16-00951-t001:** Experimental conditions.

Mode	Power Flux Density (W/cm^2^)	Pulse Duration	Frequency (GHz)	Pulse Repetition Rate (Hz)
1	8	25 ns	2.85	25
2	8	3 µs	2.85	25

**Table 2 materials-16-00951-t002:** The calculated values of the dielectric and magnetic permeabilities of Al micropowder at various frequencies.

Frequency, GHz	μ_eff_	ε_eff_
2.85	0.49 + 0.20*i*	777.80 + 0.88*i*
9.40	0.37 + 0.14*i*	777.84 + 0.94*i*

## Data Availability

Not applicable.

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
