# Peer review of "Influence of Short-Pulse Microwave Radiation on Thermochemical Properties Aluminum Micropowder"

_materials, 2023, doi:10.3390/ma16030951_

Round 1

Reviewer 1 Report

The manuscript addresses The Interaction of Aluminum Powder with Pulsed Microwave Radiations.

No direct applications are mentioned. A clear goal for the manuscript and the novelty must be highlighted at the end of the Introduction.

Results: Lack of critical discussion. They are not compared with literature. Lines 208. Is this value for oxidation temperature feasible? Reference? Other rsults should be compared with literature.

Conclusions are too broad and general. The authors should highlight the main findings and link them to the main goal of the paper. No references must be included in the conclusions, as the conclusions are independent from references. Research direction or recommendations?

Reviewer 2 Report

Please see the enclosed file.

Round 2

Reviewer 2 Report

The amendments made are satisfactory, and now the manuscript is in good shape and acceptable for publication.